# Brain Tumor Detection and Classification Using an Optimized Convolutional Neural Network

**DOI:** 10.3390/diagnostics14161714

**Published:** 2024-08-07

**Authors:** Muhammad Aamir, Abdallah Namoun, Sehrish Munir, Nasser Aljohani, Meshari Huwaytim Alanazi, Yaser Alsahafi, Faris Alotibi

**Affiliations:** 1Department of Computer Science, Sahiwal Campus, COMSATS University Islamabad, Sahiwal 57000, Pakistan; muhammadaamir@cuisahiwal.edu.pk (M.A.); sehrishjonaid@gmail.com (S.M.); 2Department of Computer Science, Superior University Lahore, Lahore 54000, Pakistan; 3AI Centre, Faculty of Computer and Information Systems, Islamic University of Madinah, Madinah 42351, Saudi Arabia; naljohani@iu.edu.sa; 4Computer Science Department, College of Sciences, Northern Border University, Arar 73213, Saudi Arabia; 5School of Information Technology, University of Jeddah, Jeddah 23218, Saudi Arabia; yaalsahafi@uj.edu.sa; 6College of Computer Science and Engineering, Taibah University, Madinah 42353, Saudi Arabia; fotibi@taibahu.edu.sa

**Keywords:** brain tumor, classification, MRI, convolutional neural network, fine-tuning, hyperparameter, detection

## Abstract

Brain tumors are a leading cause of death globally, with numerous types varying in malignancy, and only 12% of adults diagnosed with brain cancer survive beyond five years. This research introduces a hyperparametric convolutional neural network (CNN) model to identify brain tumors, with significant practical implications. By fine-tuning the hyperparameters of the CNN model, we optimize feature extraction and systematically reduce model complexity, thereby enhancing the accuracy of brain tumor diagnosis. The critical hyperparameters include batch size, layer counts, learning rate, activation functions, pooling strategies, padding, and filter size. The hyperparameter-tuned CNN model was trained on three different brain MRI datasets available at Kaggle, producing outstanding performance scores, with an average value of 97% for accuracy, precision, recall, and F1-score. Our optimized model is effective, as demonstrated by our methodical comparisons with state-of-the-art approaches. Our hyperparameter modifications enhanced the model performance and strengthened its capacity for generalization, giving medical practitioners a more accurate and effective tool for making crucial judgments regarding brain tumor diagnosis. Our model is a significant step in the right direction toward trustworthy and accurate medical diagnosis, with practical implications for improving patient outcomes.

## 1. Introduction

The human brain is the most sophisticated organ in the body, with several complex neuronal systems. These neurons are interlinked and work together with the help of instructions given by the other parts of the body. When these neurons start working abnormally, brain tumors occur. Brain tumors are the world’s second leading cause of mortality, accounting for around 9.6 million deaths in 2018. Around one out of every ten deaths worldwide occurs due to brain tumors [1]. There are two categories of brain tumors: primary and secondary. Brain tumors are recognized as malignant (cancerous) and benign (non-cancerous). The benign tumors do not spread or harm the other parts of the body, while malignant tumors spread all over the human body and can lead to life-threatening situations. Glioma, meningioma, and pituitary tumors stand out as separate forms of brain tumors [2]. Gliomas are the most common primary brain tumors and are classified into four degrees of malignancy, with higher grades signifying increased aggressiveness. These cancers begin in the glial cells of the brain [3,4]. Meningiomas are benign tumors that arise from the meninges tissue layer and are characterized by sluggish growth and limited distribution. Pituitary tumors, on the other hand, form on the pituitary gland and are generally benign and less common [5,6].

The occurrence signs of brain tumors vary depending upon the size, shape, and location of tumors. Some of the common signs of brain tumors are headache, nausea, vomiting, seizures, and difficulty in thinking or speaking [7]. The most famous brain tumor diagnosis methods include a combination of imaging tests, such as MRI and CT scans, and biopsy, depending on the type and stage of the tumor [8]. Other treatment possibilities include surgery, radiation therapy, and chemotherapy [9]. The goal of treatment is to relieve signs, control the progression of the tumor, and prolong survival.

Brain tumors affect persons of diverse categories, like ages and genders, regardless of the identification of certain risk factors. Risk factors include genetic disposition, ionizing radiation revelation, and certain hereditary illnesses. Nonetheless, the particular causes of most brain tumors remain unidentified [10]. Moreover, progressions in medical imaging and diagnostic methods have made it promising to determine earlier brain tumors, increasing the probability that interference would be effective.

Brain tumor detection and classification accuracy are critical to selecting the best treatment method. Frequently, histological investigation of tumor tissue attained from operations is used in traditional classification methods. However, in the last numerous years, there have been significant developments in the classification and diagnosis of brain tumors without surgery. These approaches comprise magnetic resonance imaging [11], CT scans [12], positron emission tomography [13], progressive neuroimaging methods known as diffusion tensor imaging [14], and magnetic resonance spectroscopy [15]. These methods provide accurate information about the tumor biography, which aids in the exact diagnosis of a brain tumor. Recent studies have focused on developing several approaches for improving the classification and detection of brain tumors. One of the most state-of-the-art developments comprises the evaluation of imaging data using artificial intelligence and machine learning algorithms [16]. These methods can significantly increase the precision of brain tumor diagnosis, classification, and patient outcome prediction [17]. In oncology, the task of precisely classifying and identifying brain tumors is difficult due to the complex and diverse structure and functional location of brain tumor tissues. Even with the prominent development in medical imaging and diagnosis, brain tumor treatment requires significant care for patients to cure them [18].

### Key Contributions of Our Work

Brain tumors constitute a major cause of cancer deaths, affecting individuals indiscriminately regardless of age or gender. Avoiding these causes requires thorough information on fundamental biological and behavioral factors due to their intricate structure and likelihood of severe neurological signs. Even though traditional diagnostic approaches are helpful, they frequently give an imprecise depiction of the extensive molecular and genetic distinctions initiated in these malignancies. The demand for sophisticated technologies that enable exact tumor characterization, leading to specific treatment plans and improved clinical findings, urges the improvement of brain tumor classification and detection approaches. This research contributes a fine-tuned CNN hyperparametric model with the following distinct characteristics:An optimized CNN hyperparameter model: The paper presents an advanced CNN hyperparameter model that has been carefully developed to optimize critical parameters in diagnosing brain tumors. The activation function, learning rate, batch, padding, filter size and numbers, and pooling layers are just a few of the carefully selected parameters that enhance the model performance and ability to generalize the model. The objective is to increase the model’s overall diagnostic accuracy and dependability by fine-tuning these hyperparameters.Datasets used: In this study, three publicly available brain MRI datasets sourced from Kaggle were utilized to test and validate the proposed model.Outstanding predictions: The proposed approach demonstrates exceptional results in average precision, recall, and f1-score values of 97% and an accuracy of 97.18% for dataset 1. These outcomes indicate the effectiveness of the optimized CNN model in accurately diagnosing brain tumors.Comparative analysis: The study extensively compares our optimized model with established techniques, affirming the strength and reliability of the findings. The proposed method consistently surpasses these approaches, showcasing its superiority in accuracy and reliability when it comes to diagnosing brain tumors.Practical implications: This model offers medical professionals a more accurate and effective tool to aid their decision-making in diagnosing brain tumors. By enhancing diagnostic accuracy and reliability, the model has the potential to advance medical imaging and improve patient care.

Our research extends our work published in [19], where we introduced a CNN hyperparameteric model for brain tumor diagnosis. The main contribution was a CNN hyperparameter model tailored for brain tumor diagnosis, fine-tuning several crucial parameters like activation functions, learning rate, batch size, number of layers, stride padding, activation functions, and filter number and size. An improved tool for assisting decision-making processes in brain tumor diagnosis is realized by the paper’s remarkable outcomes, which included high recall, accuracy, F1-score, and precision. Two datasets (containing 7023 and 253 images, respectively) [20] of various tumor types, including pituitary, glioma, meningioma, and no tumor, were used to validate the model. Additionally, the article compared the proposed model against current methods, showing superior performance results. However, our previous model suffered from several limitations. First, the model was tested on a small dataset to validate the optimized hyperparameters and assess its performance. Second, the model’s accuracy and other performance values were not optimal. Third, the model needs to be generalized to explore the integration of different imaging modalities.

The remainder of this article is organized as a related work section providing an overview of relevant research in the field, followed by a methodology section describing the overall approach employed. Subsequently, the results section explains the outcomes resulting from the applied model. Lastly, the conclusion summarizes the findings and proposes existing gaps for future research.

## 2. Related Work

Recently, significant research has been conducted to detect and classify brain tumors and assist in saving human lives. Research in [21,22] presents a brain tumor classification model based on deep learning and optimization algorithms. The proposed approach improves the efficiency and accuracy of tumor diagnosis, addressing the limitations of manual methods and providing a potential solution for automated detection approaches. Another study [23] proposes a unique CNN-based MRI brain tumor classification technique. The study compares the performance of the optimized CNN model with five pre-trained models on the CE-MRIs dataset. The model achieves excellent validation accuracy, outperforming state-of-the-art approaches, and shows the possibility of automated hyperparameter optimization in MRI brain tumor classification. CNNs outperform traditional approaches in brain tumor diagnosis by automatically learning and extracting hierarchical characteristics, resulting in increased accuracy and performance. They provide scalability, versatility, and a comprehensive learning framework, making them more efficient and effective for medical imaging analysis. The research in [24] suggests a novel hybrid deep learning model combining SqueezeNet with SVM and fine-tuning for classifying brain tumors from MRIs. The model was tested on a dataset consisting of glioma, meningioma, pituitary, and no tumors, achieving an accuracy of 96.5%. This approach outperforms existing methods, offering a precise, efficient medical diagnosis tool, highlighting significance through innovative deep learning techniques.

The work in [25] presents a thorough overview of recent improvements in automatic segmentation of white matter hyperintensities (WMHs) from brain magnetic resonance imaging, emphasizing the role of deep learning and big data in enhancing segmentation accuracy and efficiency. It analyzes several deep learning models and datasets used in the field, providing information on their effectiveness and limits. Another research paper [26] compares many local thresholding techniques for segmenting white matter hyperintensities in brain magnetic resonance imaging. The study concludes that the adaptive thresholding algorithm surpasses others in terms of segmentation accuracy and robustness, making it the most effective method for WMH segmentation.

The study in [27] explores the performance of CNN in classifying MRI scans of brain tumors. Using two MRI datasets, the study analyzes various CNN architectures, including VGG, ResNet, EfficientNet, and ConvNeXt. The best model performed well in terms of accuracy, with average precision rates of 95.3% for images without tumors, 93.8% for meningioma, 97.9% for pituitary tumors, and 94% for gliomas. The authors in [28] use the Harris Hawks optimization algorithm to optimize CNN, achieving excellent tumor recognition accuracy on a Kaggle dataset. The proposed method enhances the identification of tumor regions and hidden edge details, offering potential benefits for early diagnosis and treatment of brain tumors. A fine-tuning method in [29] uses machine learning to categorize gliomas in brain magnetic resonance imaging (MRI), focusing on reducing picture inconsistencies across different medical facilities. The advantages of this approach include improved accuracy in glioma segmentation, as evidenced by the acquired accuracy rate of 87.5% in experimental data. However, there are some limitations to consider. The fine-tuning procedure can be computationally costly, and the method performance depends on data availability and quality. Despite these limitations, the work emphasizes the potential of the proposed approach to considerably increase glioma segmentation accuracy, making it an attractive route for future research and clinical applications. Another paper [30] blends handcrafted and deep learning characteristics for brain tumor diagnosis, attaining a high accuracy of 91.2%. The method produces reliable findings, although it may necessitate significant computational resources. However, the prediction performance is impacted by balancing varied features and data quality.

The study in [31] explores brain tumor detection by combining statistical and machine learning techniques, achieving 85.6% accuracy. However, problems arise as feature choice and the dataset size can improve the method performance. Despite its limitations, the study shows that merging these methods can lead to the effective detection of brain tumors. The research in [32] examines brain tumor identification using deep learning techniques for big data processing. One benefit is a notable accuracy of 94.3%, which signifies the efficiency of neural networks in tumor identification in medical imaging research. However, deep networks require significant data labeling and substantial computing power for training.

A technique for identifying and classifying brain tumors in MRIs is proposed in [33]. However, the method cannot be used for all tumor sizes and types. Another study [34] presents a novel, trustworthy BrainMRNet model for magnetic resonance imaging-based brain tumor detection. This model applies advanced neural networks to detect tumors with high accuracy. Training deep networks is computationally expensive and efficiency may fluctuate based on the range of tumor types and imaging conditions.

The research in [35] proposes a neural network-based technique for wireless infrared imaging sensor-based brain tumor detection, with an accuracy rate of 87.2%. Drawbacks include its dependency on high-quality data and the need for an optimal sensor site. A CNN method is used in [36] for detecting brain tumors, with a notable accuracy of 93.8%. Conversely, the prerequisite for a lot of labeled data and computing resources for training is a disadvantage. The CNNs are a valuable technique for precise brain tumor diagnosis, and this work highlights their importance in medical image processing.

The study in [37] shows that brain tumors are identified with an accuracy of 95.62% by applying a maximal fuzzy sure entropy improved CNN model. Disadvantages include the complexity of merging several methods and the need for a large amount of training data. This work shows the potential of the collective approach to improve brain tumor identification and depicts its role in enhancing medical image processing. In another work [38], brain tumors in MRIs are recognized and segmented using deep learning methods, achieving a 90% accuracy. However, extensive processing resources are required for the training and inference stages. 

The study in [39] reviews how deep learning was used to identify brain tumors between 2015 and 2020, showing a notable rise in accuracy, reaching as high as 95.6%. A deep neural network correlation learning technique for classifying CT brain tumors is described in [40]. Drawbacks include the complexity of neural network architectures and their extensive computational demands.

## 3. Materials and Methods

This section explains the methodology workflow employed in this research, with detailed steps provided subsequently.

### 3.1. MRI Dataset

Our research utilized three publicly available brain tumor MRI datasets from Kaggle [20]. The details of the datasets and their divisions (i.e., training and testing) are listed in Table 1. The table presents three MRI datasets, their respective class distributions, and the number of images in the training and testing sets. Dataset 2 includes classes labeled as yes and no, with 239 images. The training set of Dataset 2 comprises 201 images, and the testing set contains 38 images, as shown in Figure 1. Dataset 3 focuses on the tumor classes, with 1500 images in total, as shown in Figure 2. The training set of Dataset 3 has 1200 images, and the testing set consists of 300 images. Dataset 3 also has an additional set of 3000 images, with 2400 in the training set and 600 in the testing set. Dataset 1 contains images of glioma, meningioma, pituitary, and no tumor classes, with a total of 7023 images, as shown in Figure 3a,b. The training set of Dataset 1 consists of 5712 images, while the testing set has 1311 images. 

### 3.2. Pre-Processing

Several processes were completed to pre-process the MRI scans for precise brain tumor analysis and insightful conclusions. This involves resolving issues with MRI images of different resolutions and intensity ranges. Normalizing intensity values improves the performance of algorithms and models. Commonly used methods include z-score normalization [41] and rescaling pixel values to the specified range (0, 1) [42]. Due to changes in position and orientation, the MRI images must be aligned to a standard reference frame. Techniques for image registration [43] make it easier to match images spatially, which improves comparison and analysis. Furthermore, it is critical to address MRI image noise and abnormalities introduced by diverse variables, such as patient movements or equipment fluctuations. Techniques for denoising the images, such as non-local means filtering or Gaussian smoothing, efficiently reduce noise without sacrificing important information [44]. 

### 3.3. Hyperparameters of CCN for Training

In convolutional neural networks (CNNs), hyperparameters [45] are parameters the user configures before training the model and are not learned during the training process as shown in Figure 4 The input layer serves as the starting point for the CNN training process, while the classification layer marks its culmination in a feed-forward manner. Conversely, the reverse procedure initiates from the classification layer and progresses through the first convolutional layer. The neuron J forwards information computed based on Equation (1) to neuron N in layer L in a forward manner. The output is determined by the non-linear ReLU function [46], as described in Equation (2).
(1)$${IP}_{N}^{L}=\sum_{J=1}^{N}W_{NJ}^{L}x_{j}+b_{N}$$
(2)$${OP}_{N}^{L}=max\left(0,{IP}_{N}^{L}\right)$$

In the given equations, IP represents the input, OP denotes the output, W stands for weight, and b represents the neuron number. Equations (1) and (2) are utilized by all neurons to generate outputs and construct the non-linear activation function based on inputs. On the other hand, pooling adopts a k-by-k filter to aggregate and compute maximum average feature values. Equation (3) demonstrates the calculation process using the SoftMax function by minimizing the new weights with the help of Equation (4), given below.
(3)$${OP}_{N}^{L}=\frac{e^{{IP}_{N}^{L}}}{\sum_{N}e^{{OP}_{k}^{L}}}$$
(4)$$C=-\frac{1}{S}\sum_{N}^{S}IP\left(X\left(\frac{a^{i}}{b^{i}}\right)\right)$$

During the training process, S represents a sample from the training set, $b^{i}$ represents the i^th^ example with its respective label $a^{i}$. The classification possibility $X\left(\frac{a^{i}}{b^{i}}\right)$ is a probabilistic measure related to the ratio of features or parameters $\left(\frac{a^{i}}{b^{i}}\right)$ for the i-th data point. This probability is used as part of a machine-learning model to improve its predictions. Using the stochastic gradient function, the model aims to iteratively update its parameters to minimize the overall cost or loss function C. In practice, this means that during each iteration of the training process, the stochastic gradient descent algorithm calculates the gradient of the cost function concerning the model parameters using only a randomly chosen subset of the data. This gradient is then used to adjust the parameters, ideally leading to a decrease in the cost function C. Over time, this process helps the model become more accurate in its classifications, as indicated by the improved classification possibility $X\left(\frac{a^{i}}{b^{i}}\right)$. Equation (5) illustrates the calculation of the weights for each convo layer known as L, with weight L having iteration t indicated as $W_{L}^{t}$.
(5)$$W_{L}^{t+1}=W_{L}^{t}+V_{L}^{t+1}$$

In the given context, $W_{L}^{t}$ represents the weight L, while $V_{L}^{t+1}$ represents the updated weight value at iteration t. This plays a critical role in feature extraction within convolutional neural networks. It consists of multiple filters that extract features from input data. Equations (6) and (7) are employed to evaluate the resulting value and layer sizes. Here, $n_{L}^{i}$ denotes the resulting feature map for images, σ represents the activation function, $y_{L}$ signifies the input width, and $x_{L}^{i\epsilon f},z^{i\epsilon f}$ are filter channels within the filter (f).
(6)$$n_{L}^{i}=\sigma \left(x_{L}^{i}-y_{L}+z^{i}\right)$$
(7)$$Convo\;layer\;ouput\;size=\left(\frac{input-size\;of\;filter}{stride}\right)+1$$

In convolutional neural networks, a pooling layer is expected to follow each convolutional layer. This layer plays a role in parameter management and helps prevent overfitting. Among the various types of pooling layers, such as min and average, max pooling [47] is the most commonly employed. The output and size for these layers are computed using Equations (8) and (9). The equation ${Pool}_{i,j}={max}_{r,s\epsilon R}$ describes the max pooling operation where the value at the (*i*,*j*)-th position of the output feature map is determined by taking the maximum value from a specific region *R* of the input feature map. This region *R* consists of all positions (*r*,*s*) within the pooling window centered around the (*i*,*j*)-th position.
(8)$${Pool}_{i,j}={max}_{r,s\epsilon R}$$
(9)$$Pooling\;layer\;size=\left(\frac{convo\;output-Pooling\;Size}{stride}\right)+1$$

### 3.4. Hyperparametric Fine-Tuning of CNN

Table 2 illustrates the importance of improving many parts of network architecture to fine-tune the hyperparameters. The number and size of filters play a crucial role in determining the depth of the convolutional layers and the receptive field. It is beneficial to explore alternative filter numbers and different filter sizes while also adjusting the stride to control the dimensions of the output feature maps. A large stride leads to a decrease in spatial resolution, while a smaller stride increases it. Appropriate padding parameters are chosen to preserve the spatial size of the input volume during convolutional processes. Padding helps mitigate the impact of boundaries while retaining essential spatial information. The user manually sets several hyperparameters to control the structure and training process of the network. These hyperparameters include the learning rate, batch size, number of epochs, optimizer, learning rate schedule, shuffle, verbose, dropout rate, filters, filter size, and activation function.

On the other hand, the machine learning process automatically optimizes certain variables, primarily the weights and biases of the convolutional and fully connected layers. These parameters are adjusted during training to minimize the loss function and improve the network’s performance on the given task. The optimization algorithm, such as stochastic gradient descent (SGD) or Adam, iteratively updates these weights and biases based on the gradients computed from the loss function.

Different pooling methods were utilized during the experiments, such as max, min, and average pooling. These methods assist in reducing spatial sizes. In a CNN, suitable activation functions known as ReLU, sigmoid, and tanh introduced non-linearity and assisted the system in developing complex relationships with inputs [48]. The choice of learning rate influences the convergence of the model. The lower learning rate may need further iterations and can lead to better convergence, and the higher one may result in more rapid convergence but with the risk of passing the optimal solution. In experiments, the batch sizes determine the number of training examples processed in each iteration. The number of fully connected and convolutional layers can be changed to change the depth of the network. Overfitting is reduced, and generalization is improved by using regularization techniques like L2 regularization or dropout. Weight updates during training are regulated by experimenting with optimization approaches such as Adam and stochastic gradient descent (SGD).

### 3.5. Working of Hyperparameteric CNN

The hyperparameterized CNN model uses hyperparameters to adjust its design and improve performance. The hyperparameter configuration for the proposed CNN model in the table appears to be a well-designed and consistent starting point for training on the three different datasets. The learning rate is set to a low value of 0.00001, the model is trained on a GPU environment, and the batch size is 8 with 50 training epochs for Datasets 2 and 3 and 8 epochs for Dataset 1. Both SGD and Adam optimizers are utilized, and the learning rate schedule follows a piece-wise approach, with the training data shuffled after each epoch. The model incorporates a 20% dropout rate and uses various filters (ranging from 2 to 128) with 3 × 3 and 5 × 5 filter sizes and the commonly used ReLU activation function. This hyperparameter configuration provides a solid foundation for the proposed CNN model and can be further refined through additional experimentation and tuning to achieve optimal performance on specific datasets. The particular configuration of the model hyperparameters is provided in Table 3.

## 4. Results

The CNN was implemented in Python on the system, which had an 8GB GTX 1060 GPU, a 12th generation i9 processor, and 32GB of RAM. This setup was utilized to predict brain tumors. The execution is available for other researchers at GitHub (https://github.com/muhammadaamir1234/Brain-tumor-hyperparameter-CNN (accessed on 11 June 2024)). 

### 4.1. Evaluation Criteria

To evaluate the proposed model’s performance in detecting and classifying brain tumors, several performance equations (i.e., (10)–(13)) were employed. True positive images, denoted as *Tp*, referred to correctly classified cases, while true negative images, denoted as *Tn*, represented correctly classified cases as negative. The quantity of incorrectly positive classified images was signified as *Fp*, and incorrectly negative classified images were denoted as Fn. Statistical metrics were calculated with the help of Equations (10)–(13) given below.
(10)$$Precision\;(Pre)=\frac{Tp}{Tp+Fp}\;$$
(11)$$Recall\;\left(R\right)=\frac{Tp}{Tp+Fn}$$
(12)$$F1-score\;\left(F1-S\right)=\frac{2\left(R*Pre\right)}{R+Pre}$$
(13)$$Accuracy\;\left(Acc\right)=\frac{Tp+Tn}{Tn+Tp+Fp+Fn}$$

### 4.2. Applied Model Results

The given classification confusion matrix represents model performance on the test dataset, displayed in Figure 5, Figure 6 and Figure 7 for Datasets 1, 2, and 3, respectively. Four separate categories make up this dataset: glioma, pituitary, meningioma, and no tumor for Datasets 1 and 2 as a yes and no for Datasets 1 and 2. Regarding the number of images used for classification, the confusion matrix (CM) is the square matrix whose proportions are equivalent to the number of classes. It is a 4 × 4 matrix in this instance because there are four classes. Actual and anticipated class labels are combined to form each cell in the matrix. Every element within the matrix represents the number of photos associated with a particular class (rows) and anticipated to be a part of a particular forecasted class. The confusion matrix in Figure 5 shows a classification model with 97.18% accuracy, correctly predicting 1274 out of 1311 instances. The model performs best on no tumor cases (100% accuracy) but has minor misclassifications between glioma, meningioma, and pituitary cases. The CM in Figure 6 shows the classification results for two classes, “Yes” and “No”, with the model achieving 93.33% overall accuracy. The model correctly predicted 18 “Yes” instances (60.00%) and 10 “No” instances (33.33%), with minor misclassifications resulting in 2 “Yes” instances and 2 “No” instances being incorrectly labeled. On the other hand, CM in Figure 7 shows the performance of a binary classification model with an overall accuracy of 96.49%. The model correctly predicted 302 “Yes” instances (50.50%) and 275 “No” instances (45.99%), with minor misclassifications resulting in 9 “Yes” instances and 12 “No” instances being incorrectly labeled. Additionally, Table 4 presents statistical data, including average precision, recall, and f1-score as 0.97 for each class.

Figure 8, Figure 10 and Figure 12 depict the training accuracy of Datasets 1, 2, and 3, respectively. It represents the proportion of correctly identified examples within the training dataset. It serves as a measure of how effectively the model absorbs and learns from the training data. During each training iteration, also known as an epoch, the model parameters are adjusted to better fit the training data, resulting in improved training accuracy.

The training loss illustrated in Figure 9, Figure 11 and Figure 13 for Datasets 1, 2, and 3 computes the variance between the models’ predicted outputs and the actual targets. The objective is to decrease the training loss of the model and adapt the data. Loss functions such as categorical cross-entropy are typically used to compute the training loss. Validation and training accuracy (Figure 5, Figure 7 and Figure 9), which were ascertained to not be used during the training phase, are used to evaluate the model’s capacity to generalize to new, unknown data. The x-axis displays the number of epochs, while the y-axis displays accuracy. The orange line represents the validation accuracy, the blue line represents the training accuracy. The graph is made up of these two lines. The validation loss (Figure 9, Figure 11 and Figure 13) means the deviation between the model predictions and the actual targets in the validation dataset. It serves as a gauge for the model performance on unseen data.

**Figure 8 diagnostics-14-01714-f008:**
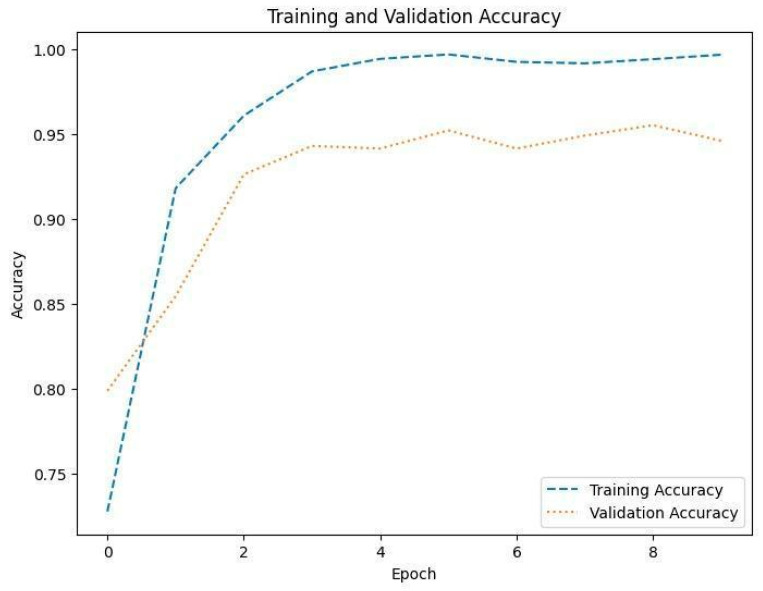
Accuracy of training and validation for Dataset 1.

**Figure 9 diagnostics-14-01714-f009:**
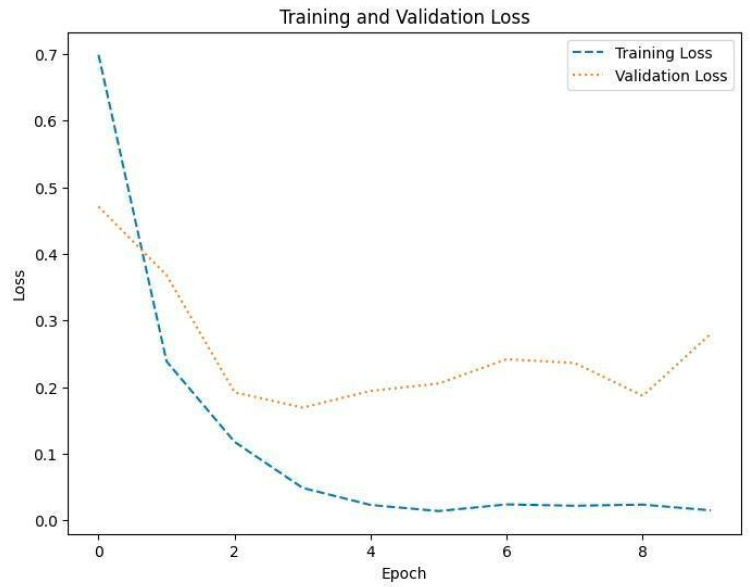
Training and validation loss in Dataset 1.

**Figure 10 diagnostics-14-01714-f010:**
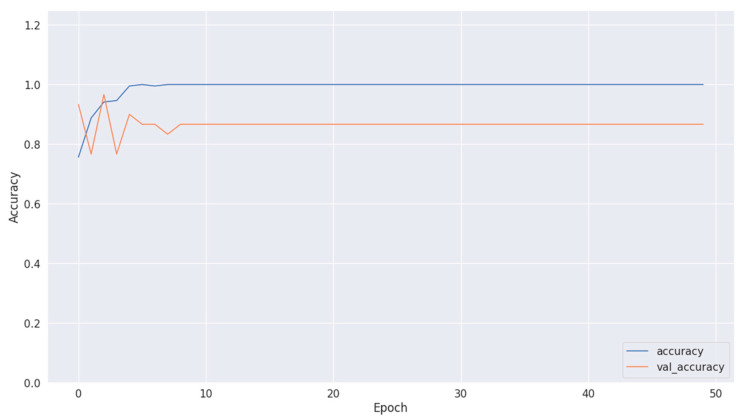
Accuracy of training and validation using Dataset 2.

**Figure 11 diagnostics-14-01714-f011:**
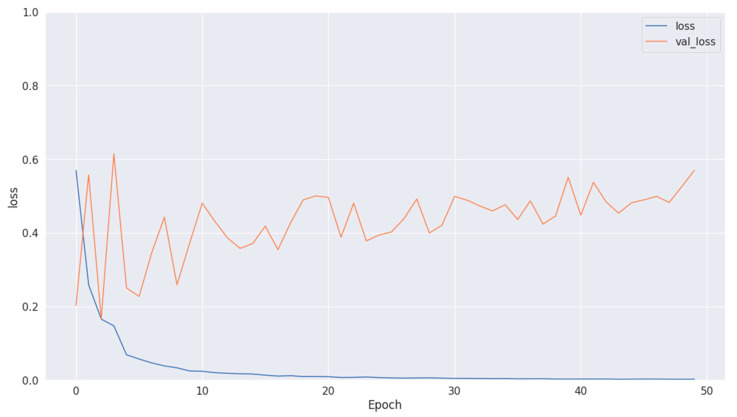
Training and validation loss in Dataset 2.

**Figure 12 diagnostics-14-01714-f012:**
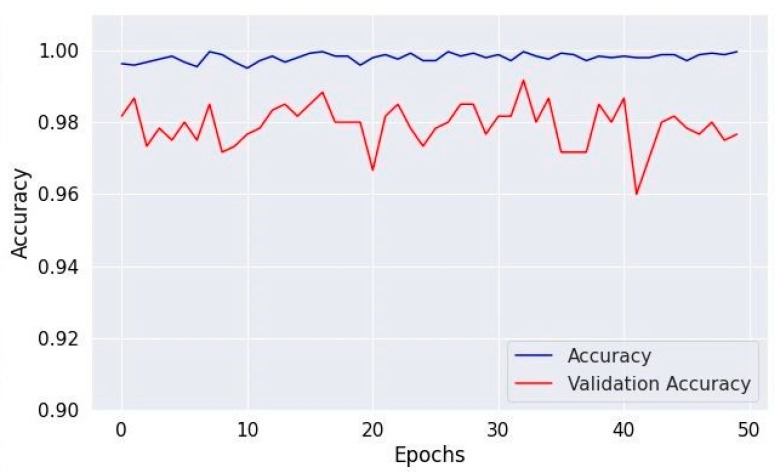
Accuracy of validation and training using Dataset 3.

**Figure 13 diagnostics-14-01714-f013:**
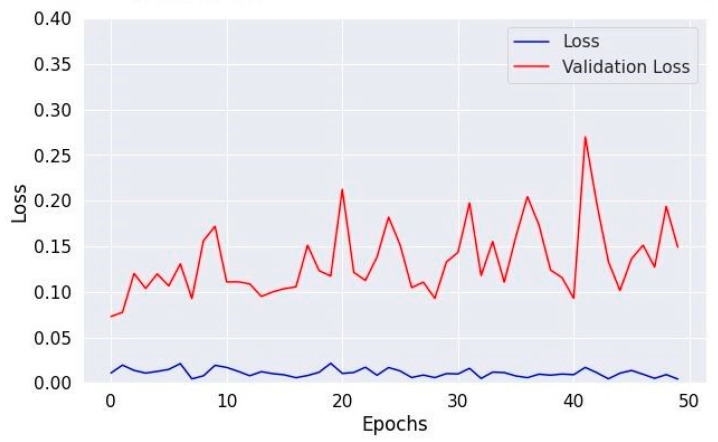
Losses in training and validation for Dataset 3.

## 5. Discussion

Table 4 provides a complete summary of the performance achieved in this work. The results presented in the table are highly impressive, showing an average statistical value of 0.97 for Dataset 1 for each term, and the overall accuracy of Dataset 1 is 97%. In Table 5, a comparison is made between the proposed techniques and state-of-the-art methods. By way of contrast, the effectiveness and consistency of the model in correctly detecting and categorizing brain tumors are illustrated. The model performs admirably and yields reliable results, as demonstrated by the statistics in these tables.

Table 5 compares different model performances in classifying brain MRI data. Our model demonstrated superior performance with an accuracy of 97%, with the same statistical values as 97% on Dataset 1. Our model produced results of 0.93 for accuracy, 0.95 for precision, 0.91 for recall, and 0.93 for f1-score on Dataset 2. Using the same statistical values as 96%, our model produced an accuracy of 0.96 on Dataset 3. These numerical values highlight the strong performance of our model compared to the other models, indicating its effectiveness in accurately classifying brain MRI data across multiple datasets. Our model provides the most reliable and balanced performance across accuracy and other statistical values for Dataset 1. This comparison demonstrates “our model” effectiveness and reliability in medical image analysis, demonstrating its potential to improve diagnostic accuracy and patient outcomes.

## 6. Limitations of the Model and Future Work

Although the CNN model employed for brain tumor detection demonstrates high accuracy and effectiveness, its limitations may involve potential overfitting due to its complexity and the need for specific tuning of hyperparameters to the dataset. This might reduce the model’s generalizability to new, unseen data or medical imaging datasets. Additionally, the reliance on a large and diverse dataset for training may not account for all possible variations of brain tumors, which could impact the model’s applicability to real-world diagnostic scenarios where data variability is high. Furthermore, the computational resources required for training and optimizing such a model could be substantial, limiting its accessibility for some medical facilities. This implies that while the model demonstrates impressive performance metrics, its effectiveness might diminish when applied to other datasets with different brain tumor characteristics, distributions, or types. This limitation emphasizes the need for further validation and testing across a broader range of datasets to ensure the model’s robustness and applicability to diverse medical imaging contexts in the future.

## 7. Conclusions

A fine-tuned CNN hyperparametric model is proposed in this research. This model improves brain tumor diagnosis accuracy by optimizing the fine-tuning parameters of the CNN model. To this end, we employed a readily accessible dataset of brain tumor MRI scans. Our method proved highly effective, boasting an accuracy of 97%, indicating discrimination abilities, primarily attributed to the meticulous adjustment of CNN hyperparameters. Significant results, including an average precision, recall, and f1-score of 97%, demonstrate excellent accuracy in detecting brain tumors. Moreover, the statistical comparison with other recent models shows that our technique performs better using the same MRI dataset. The proposed method may serve as a guideline and assist in the future medical field for brain tumor detection and classification.

The implications of the hyperparametric CNN model used for brain tumor detection are significant and transformative for medical diagnostics. The model receives a high accuracy of 97% by methodically adjusting important hyperparameters, including the number of filters, filter sizes, and learning rate. Our optimization technique makes the diagnosis method more reliable and accurate for brain tumors, giving medical practitioners a more effective and accurate tool. The practical implications of such a model are profound, with the potential to improve patient care by offering more precise and reliable diagnostic information that supports critical decision-making processes in treating and managing brain tumors.

## Figures and Tables

**Figure 1 diagnostics-14-01714-f001:**
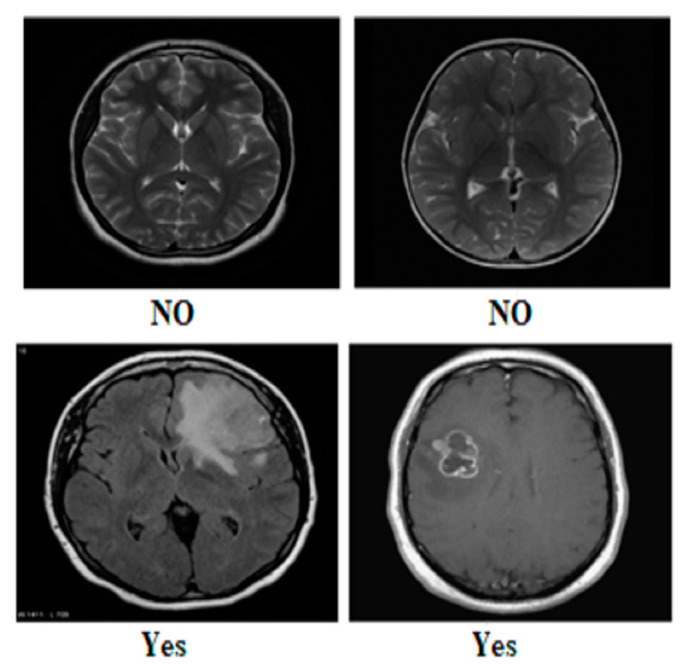
Sample images from Dataset 2.

**Figure 2 diagnostics-14-01714-f002:**
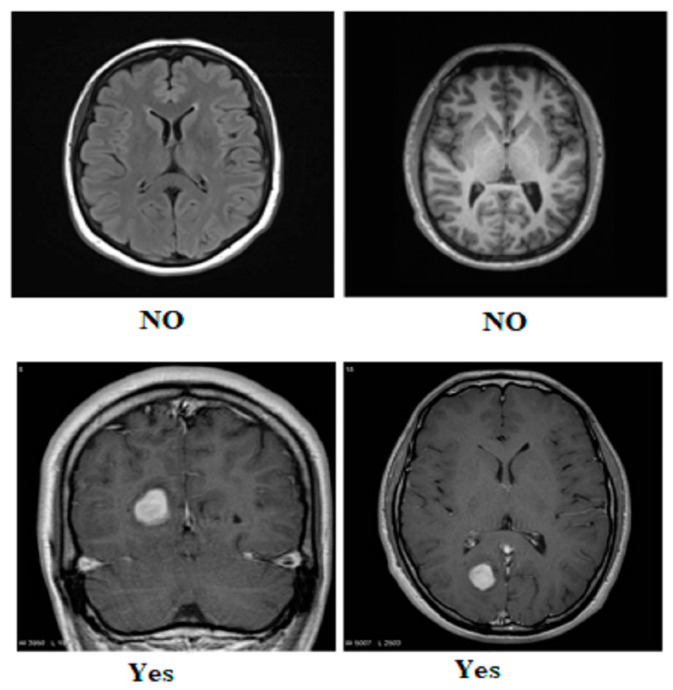
Sample images from Dataset 3.

**Figure 3 diagnostics-14-01714-f003:**
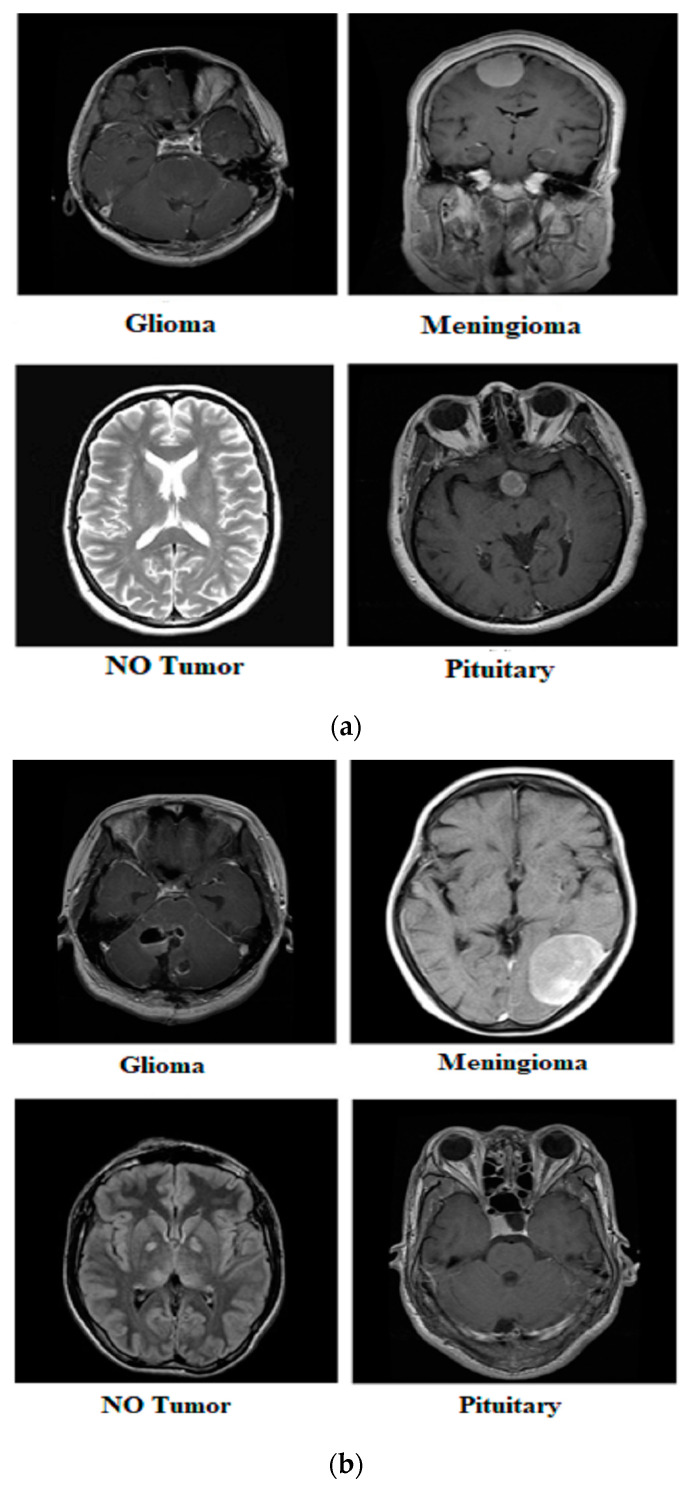
Samples from Dataset 1.

**Figure 4 diagnostics-14-01714-f004:**
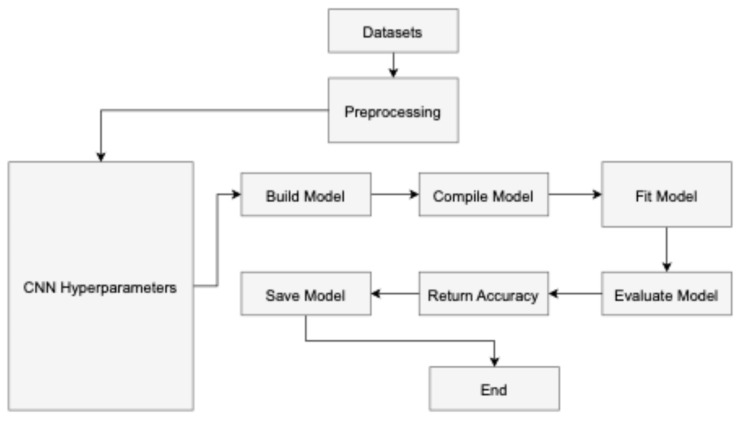
The workflow of the proposed model.

**Figure 5 diagnostics-14-01714-f005:**
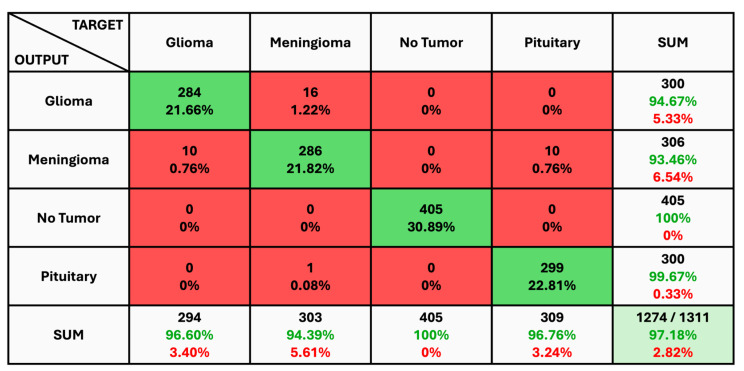
The Confusion Matrix for Dataset 1.

**Figure 6 diagnostics-14-01714-f006:**
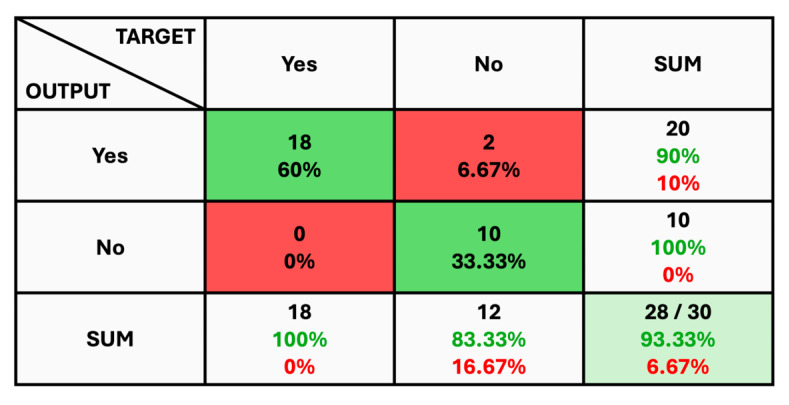
The Confusion Matrix for Dataset 2.

**Figure 7 diagnostics-14-01714-f007:**
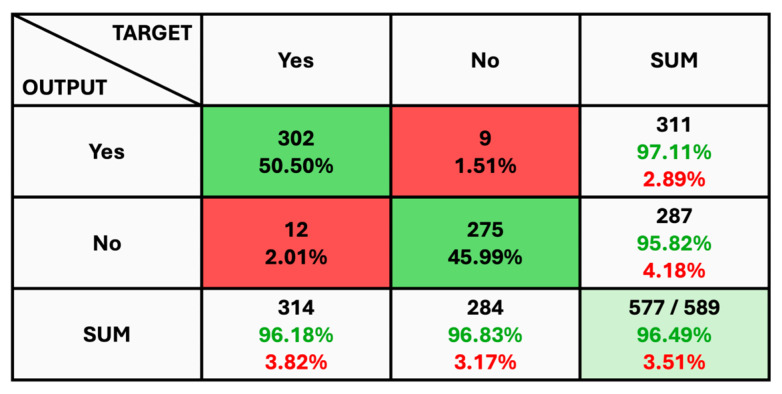
The Confusion Matrix for Dataset 3.

**Table 1 diagnostics-14-01714-t001:** Brain tumor MRI dataset.

Dataset 1				Dataset 2				Dataset 3			
Class	Images	Train	Test	Class	Images	Train	Test	Class	Images	Train	Test
Glioma	1621	1321	300	Yes	155	135	20	Yes	1500	1200	300
Meningioma	1645	1339	306	No	84	66	18	No	1500	1200	300
Pituitary	1757	1457	300								
No Tumor	2000	1595	405								
**Total**	**7023**	**5712**	**1311**	**Total**	**239**	**201**	**38**	**Total**	**3000**	**2400**	**600**

**Table 2 diagnostics-14-01714-t002:** Steps for fine-tuning of CNN.

Fine-Tuning of CNN Hyperparameter
**Step 1:** Find the best hyperparameters to train the final model. **Step 2:** Develop new model instances for the best hyperparameters. **Step 3:** Train the model with the specified parameters. **Step 4:** Test and evaluate the CNN model. **Step 5:** Find the best performance metrics (e.g., accuracy).

**Table 3 diagnostics-14-01714-t003:** Hyperparameter configuration for proposed CNN.

Sr. No	Parameters	Dataset1	Dataset2	Dataset3
		Values	Values	Values
1	Batch size	8	8	8
2	Epochs	8	50	50
3	Optimizer	SGD, Adam	SGD, Adam	SGD, Adam
4	Learning rate	Epochs 0–5, Learning Rate: 0.0001 Epochs 6–8, Learning Rate: 0.00005 Epochs 9–10, Learning Rate: 0.00001	Epochs 0–20, Learning Rate: 0.0001 Epochs 21–30, Learning Rate: 0.00005 Epochs 31–50, Learning Rate: 0.00001	Epochs 0–20, Learning Rate: 0.0001 Epochs 21–30, Learning Rate: 0.00005 Epochs 31–50, Learning Rate: 0.00001
5	Shuffle	Every epoch	Every epoch	Every epoch
6	Dropout rate	0.2	0.2	0.2
7	Number of filters	16, 32, 64, 128	2, 4, 16, 32, 64	4, 8, 16, 32, 64
8	Filter size	3 × 3, 5 × 5	3 × 3, 5 × 5	3 × 3, 5 × 5
9	Activation function	ReLU	ReLU	ReLU

**Table 4 diagnostics-14-01714-t004:** Results of model performance.

Dataset 1					Dataset 2					Dataset 3				
Class	Pre	R	F1-S	Acc	Class	Pre	R	F1-S	Acc	Class	Pre	R	F1-S	Acc
Glioma	0.95	0.97	0.96	97.18	Yes	0.90	1.00	0.95	0.93	Yes	0.97	0.96	0.97	0.96
Meningioma	0.93	0.94	0.94		No	1.00	0.83	0.91		No	0.96	0.97	0.96	
No Tumor	1.00	1.00	1.00											
Pituitary	1.00	0.97	0.98											
Average	0.97	0.97	0.97			0.95	0.91	0.93			0.96	0.96	0.96	

**Table 5 diagnostics-14-01714-t005:** Results comparison using the current techniques.

Method	Dataset	Acc	Pre	R	F1-S
Inception-V3 Fine-tuned model [49]	Brain MRI	0.94	0.93	0.95	0.94
MobileNetV2 [50]	Brain MRI	0.92	0.93	0.90	0.91
Deep-Net: Fine-Tuned model [51]	Brain MRI	0.95	0.93	0.94	0.95
CNN model [19]	Brain MRI Dataset1 Dataset 2	0.96 0.88	0.94 0.87	0.94 0.87	0.94 0.87
**Our model**					
**Dataset 1**	Brain MRI	0.97	0.97	0.97	0.97
**Dataset 2**	Brain MRI	0.93	0.95	0.91	0.93
**Dataset 3**	Brain MRI	0.96	0.96	0.96	0.96

## Data Availability

The data collected and analyzed during this study are available from the corresponding author upon reasonable request.

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
