# Peer review of "Brain Tumor Detection and Classification Using an Optimized Convolutional Neural Network"

_diagnostics, 2024, doi:10.3390/diagnostics14161714_

Round 1
Reviewer 1 Report
Comments and Suggestions for Authors
1. Compared with the content of "Table 1", the description of "Dataset 3" in this paper is difficult for readers to understand. The authors need to clearly state the label type of "Dataset 3" (i.e. binary or multi-class classification; in line 219 of the manuscript, it said "Dataset 3 focuses on the tumor class..."), the amount of data (i.e. 1500 vs 3000), and group distribution (i.e. train vs test).
2. In section "3.3. Hyper-parameters of CCNs (should it be CNN?) for Training", the authors describe the main concepts of CNN. However, they did not explain clearly the meaning of some variables (for example: cost C and classification possibility X in formula (4); j and r in formula (8)). Furthermore, they did not specify which variables were hyper-parameters fine-tuned by the user and which variables were automatically optimized by the machine learning process.
3. In Table 3, the authors list the fine-tuning settings of hyper-parameters. However, the hardware environment in which the classification models were trained (i.e. executable environment GPU) and the log information printed during the training procedure (i.e. verbose) do not belong to hyper-parameters. Conversely, what are the arguments (i.e. boundaries and interval values) for the piece-wise schedule of learning rate?
4. Section "3.6. Strengths of the proposed model" is a redundant paragraph that should be deleted. The main contributions of this paper should be explained in the introduction, discussion, and conclusion paragraphs.
5. In "4. Discussion" of this paper, the description of the definition of confusion matrix is redundant and should be deleted. Instead, the authors should elaborate on the meaning of the percentage values represented in Figures 5, 6, and 7.
6. In "5. Discussion" of this paper, the authors claim that the prediction model constructed by the proposed method achieves better performance than the current research results. However, they appear to be directly quoting the performance values from the reference literatures. Such comparisons are meaningless if identical training dataset is not used to construct the predictive models. The quality and quantity of different training datasets have a significant impact on the predictive performance of the constructed model. Therefore, the authors had to confirm that these references and their team used identical training dataset before they could compare the performance differences between the prediction results. Or the research team needs to use the respective methods of these references to build the prediction model with Dataset 1, 2, or 3 of this paper. Then the performance differences between the testing results could be compared.
7. The chapters "7. Implications of the model" and "8. Conclusion" of this paper should be merged. There is no need to forcefully divide it into two chapters.
Comments on the Quality of English Language
This paper is rife with numerous editing errors.
1. Why not set the number of figures according to the order of the text?
2. In this paper, the chapter titles, table headers, and figures headers are sometimes all capitalized, and sometimes the first letter is capitalized.
3. The sample images of "Figure 1" and the content of "Table 5" are divided by pages.
4. For "Figure 8~13", are these line charts showing accuracy and loss values generated using different softwares? Why not use a unified format?
5. In line 345 of this paper, it said "...while true negative images, denoted as Tn, represented incorrectly classified cases as negative". This does not fit the usual definition of a true negative. Also, in line 385, it said "The validation loss (Figure 6, 8, 10)...". However, the content of these figures does not provide any validation loss values.
6. The writing of this paper requires consulting professional English editing services.
Reviewer 2 Report
Comments and Suggestions for Authors
Round 2
Reviewer 2 Report
Comments and Suggestions for Authors
The work that has been done is great.
Author Response
Thank you for valuable comments and suggestions for our work.